# Effect of Aggregate Type on Properties of Ultra-High-Strength Concrete

**DOI:** 10.3390/ma15145072

**Published:** 2022-07-21

**Authors:** Anna Szcześniak, Jarosław Siwiński, Adam Stolarski

**Affiliations:** Faculty of Civil Engineering and Geodesy, Military University of Technology, 2 Gen. S. Kaliskiego Street, 00-908 Warsaw, Poland; jaroslaw.siwinski@wat.edu.pl (J.S.); adam.stolarski@wat.edu.pl (A.S.)

**Keywords:** ultra-high-strength concrete, mechanical properties, natural fine aggregates

## Abstract

In this work, we present an analysis of natural fine aggregates’ influence on the properties of ultra-high-strength concrete. The reference concrete mix was made of natural sand with the addition of fly ash and microsilica. It was assumed to obtain concrete with a very high strength without the addition of fibers and without special curing conditions, ensuring the required workability of the concrete mix corresponding to the consistency of class S3. The reference concrete mix was modified by replacing sand with granite and basalt aggregate in the same fractions. Five series of concrete mixes made with CEM I 52.5R cement were tested. Experimental investigations were carried out regarding the consistency of the concrete mix, the compressive strength, the flexural strength and the water absorption by hardened concrete. A comparative analysis of the obtained results indicated significant improvement in the concrete strength after the use of basalt aggregate. The strength of the concrete series based on basalt aggregate, BC1, allowed it to be classified as ultra-high-performance concrete. Concrete based on sand, SC1, was characterized by the lowest compressive and flexural strength but obtained the best workability of the mix and the lowest water absorption. The results presented in the paper, show a significant influence of the type of aggregate used on the mechanical and physical properties of ultra-high strength concrete.

## 1. Introduction

High-performance concretes (HPCs) and ultra-high-strength concretes (UHSCs) are predominantly used to produce small architectural elements such as facades, etc. However, these materials are increasingly being used in large volumes to construct structural elements of bridges, tunnels, runways, etc. [1]. The use of ultra-high-performance concretes (UHPCs) allows for a significant reduction in the dimensions and weights of construction elements and provides the freedom to create even the most futuristic construction solutions. In addition, UHPCs are characterized by increased durability and resistance to aggressive external factors [2]. The use of additives, such as microsilica, which has abundant pozzolanic properties, increases the tightness and strength and reduces the porosity and swelling of the concrete. UHPCs are characterized by increased resistance to freeze/thaw, chloride penetration and abrasion.

The design and analysis of the mechanical and physical properties of HSCs and UHPCs are the subjects of various studies. A significant portion of these studies concerns concretes made with the use of steel or composite fibers. Kusumawardaningsih et al. in [3,4] showed that the addition of fibers increases the tensile strength of UHPCs and decreases its compressive strength. In [5], Smarzewski showed the influence of steel, polypropylene, basalt and glass fibers on the mechanical properties and consistency of high-performance cementitious composites. The highest compressive and splitting tensile strength was obtained by using steel fibers with 50 mm length and hoked ends, while the highest flexural strength was obtained by using glass, straight fibers with 18 mm length. A comparison of the effect of high-strength steel fibers addition on the mechanical properties of traditional concrete and HPC was presented by Máca et al. in [6]. The addition of fibers reduced the compressive strength of traditional concrete and increased its flexural strength. In the HPC, the addition of fibers increased both, the compressive and flexural strength.

The use of UHPCs is limited due to the technological difficulties in their production and the high costs, especially of the constituent materials. As Zhao and Bo showed in [7], casting positions significantly influence the mechanical properties of UHPC specimens. The effect of various curing conditions on the early- age strength of UHPCs was presented by Park et al. in [8]. Their experimental results indicated that temperature curing conditions during the early-age significantly accelerate the strength of concrete. Drzymała et al. in [9] showed that curing at a temperature of even 300 °C increases the compressive strength of high-performance air-entrained concrete. Abokifa and Moustafa in [10] pointed out the necessity of reducing UHPC manufacturing costs and proposed the use of locally available materials for the production of precast bridge deck systems. Some studies have also been conducted on the use of waste materials to produce UHPCs. Such a solution allows one to reduce costs through waste management, and it is ecological, which is highly valued today. Ambily et al. in [11] noted the possibility of applying copper slag as fine aggregate in UHPCs. The test results showed that the application of copper slag in the manufacturing of UHPCs with fibers is helpful, but it requires particular curing conditions, including heat treatment. In [12], Kumar et al. argued that recycled concrete aggregates in amounts of up to 20% can be used as a replacement of natural aggregates for HPC production without having a negative influence on its physical and mechanical properties. Ali et al. [13] indicated the usefulness of applying waste glass powder instead of silica fumes in the fabrication of HSC exposed to high temperatures reduces the degradation of its mechanical properties. Zhao and Sun [14] conducted an in-depth analysis of the nano-mechanical behavior of a so-called green UHPC which contained utilized river sand and fly ash. The test results showed that a significant portions of the fly ash and cement remained unreacted in the hydration processes. However, these unreacted particles had more abundant mechanical properties than the hydration products. The authors concluded that unreacted fly ash and cement functioned as micro-aggregates and strengthened the UHPC paste. Waste materials can also be used as fibers in concrete and cement composites [15], increasing their strength properties. As Małek et al. showed in [16], the addition of metal lathes waste from postproduction of mechanical components increases the mechanical properties of concrete. Sobuz et al. [17] emphasized the advisability of producing UHPCs with the use of commonly available materials and without special curing conditions. The test results indicated that, for the adopted assumptions, the concrete compressive strength of over 150 MPa was obtained using pit sand as a fine aggregate and steel fibers. However, among 40 tested mixtures, after 28 days of curing, only one of the compressive strengths of concrete exceeded 150 MPa.

At the beginning of the 20th century, Donza et al. in [18] conducted research on the influence of the type of fine aggregate on the properties of high-strength concrete. Concretes based on natural sand, granite, limestone and dolomite were analyzed. These tests were carried out with a long maturation time, but the compressive strength of the concrete did not exceed 80 MPa. A comparative analysis of high strength concrete (HSC) based on andesite and granite aggregates in terms of strength and elastic modulus was presented by Beushausen and Dittmer in [19]. The tests carried out using concretes with compressive strengths in the designed range of 30–120 MPa showed that concrete based on granite aggregate had higher compressive strength and a lower elastic modulus compared to andesite concrete. Basalt is a magma effusive rock with a content of 45–52% SiO_2_, which has a compressive strength of 150–300 MPa and a tensile strength of 10–20 MPa. Experimental investigations on the use of basalt aggregate in high-strength concrete mixes, which were conducted by Kishore et al. and presented in [20], indicate its influence on concrete strength improvement and reductions in water demand in a mix. However, these studies focused on concrete mixtures with compressive strengths of 90 MPa. In [21], Li et al. showed that in UHPCs, the addition of coarse basalt aggregate leads to a decrease in mechanical strength. An analysis of a range of mixes’ designs and a performance evaluation of UHPC with basalt aggregate is presented in [22]. The obtained results also showed that the increase in the particle size of coarse basalt aggregate led to a decrease in the mechanical strength of UHPC. Experimental tests regarding UHPCs with basalt aggregate are presented in [2]. In all cases, the use of basalt aggregate in UHPCs has been evaluated positively.

Ultra-high-strength concrete mixes are characterized by a very high density, which significantly reduces their workability. Chemical admixtures partially overcome this problem, but have limited uses. A dense consistency makes it difficult or even impossible to use concrete in elements with a complex reinforcement structure. Complex internal structures usually require the use of modified cement mortar, instead of concrete. An example of such a solution was presented by Małek et al. in [23]. The use of fly ash additive was shown to improve the workability of a concrete mix and its tightness and resistance to the penetration of aggressive external factors. Still, it simultaneously reduced the early strength of concrete [24]. In [25] Golewski and Szostak proposed the application of a specifically formulated chemical nano-admixture in the grains form of C-S-H phase particles to increase the early strength of concretes with the addition of fly ash.

This paper presents the results of research devoted to the development of a concrete recipe that enables high and very-high-strength concrete to be obtained while maintaining the required consistency of the concrete mix. The analyzed concrete mixes were formulated with the intention to be used in the manufacturing of concrete elements containing spatial reinforcement, with a high density of reinforcement bars and small sized reinforcement meshes, which are the subject of the patent description [26]. Therefore, in these types of mixes, it is imperative to maintain the consistency that enables the proper filling and densification of concrete in an element with a very complex reinforcement structure. The consistency of class S3 was measured using the slump test, which was used as required. It was also assumed that concrete should be made of typical, available materials without the use of fibers and special curing conditions. The analysis was carried out with concretes based on sand, granite and basalt as fine aggregate with the addition of fly ash and microsilica, and then, their physical and mechanical properties were compared. 

Tightness is a characteristic feature of UHSCs, which increases the durability of the structure. Therefore, in the paper a test which allows for a preliminary assessment of concrete resistance to water penetration under the influence of atmospheric pressure was performed. Studies with similar scopes were presented by Golda and Kaszuba [27] and by De Schutter and Audenaert [28]. In the presented method, the mass of samples exposed to water was related to the mass of hardened concrete, to better reflect the real working conditions of concrete.

## 2. Materials

The tests were carried out for five series of concrete mixes using CEM I 52.5 R Portland cement and fine aggregate to 2 mm. The series of concrete mixes differed in the type of aggregate and the amount of cement that was in them. One series of samples was prepared based on pit sand, two based on granite aggregate and the other two based on basalt aggregate.

The materials used for the tests are as follows:White Portland cement CEM I 52.5R (c);Fine aggregate—pit sand, granite, and basalt with fractions of 0.125–0.25 mm, 0.25–0.5 mm, 0.5–1.0 mm and 1–2 mm;Fly ash (fa);Microsilica (ms);Polycarboxylate admixture;Pure laboratory pipeline water (w).

Cement CEM I 52.5 R with one of the highest compressive strengths available on the market was used to the tests. The compressive strength of cement after 2 days of maturation is 44 MPa and after 28 days it is 72 MPa. In each of the tested series of concrete, the same fly ash additive was used to ensure an increase in the workability of the concrete mixes. The fly ash used was characterized by fineness below 12% and, loss on ignition below 5% and was certified according to the EN 450–1 standard [29]. The detailed chemical composition of fly ash is presented in Table 1 [24]. Similar to fly ash, the same amount of microsilica was used in each series of concrete mixes. The chemical composition of the cement and microsilica used were determined using X-ray fluorescence [30] (Table 1).

## 3. Methods

### 3.1. Mix Proportions

To investigate the influence of the type of aggregate on the physical and mechanical properties of high-strength concrete, first the composition of the primary SC1 mix based on sand was developed. The main aim during the design of the SC1 mix was to obtain high strength parameters while maintaining the required consistency class of the mix. The required consistency class was determined as plastic, which corresponded to class S3 according to the slump test. Class S3 is defined as the consistency for which the cone fall is between 100 and 150 mm. Then the primary mix was modified by replacing sand with granite and basalt aggregate The GC1 and BC2 concrete mixes are equivalents of SC1 mix made on granite and basalt, respectively, including the volume density of the aggregates. Moreover, two additional concrete mixes GC2, and BC1 were analyzed. The GC2 concrete composition was correlated with the proportion of BC2 concrete, and the BC1 concrete composition was correlated with the GC1 concrete composition in a similar way. The difference in the amount of cement used in relation to 1 m^3^ for concretes BC1 and BC2 as well as GC1 and GC2 is the same, i.e., 6%. All of the recipes maintained the same water–cement ratio w/c = 0.33, water–binder ratio w/b = 0.26 and the consistency class S3. Moreover, to improve the workability of the concrete mixes, fly ash was used in proportion of fa/c = 0.167. Microsilica in proportion of ms/c = 13.3 was used to improve the tightness of concrete and increases its strength. The concrete mixes’ compositions are given in Table 2.

### 3.2. Concrete Mix Consistency Testing

The slump test following EN 12350–2 [31] was used to determine the consistency of the concrete mixes. The test was carried out in laboratory for fresh concrete for the SC1, GC1, GC2, BC1, BC2 series immediately after making the concrete mix, in a time not exceeding 8 min. The cone was filling with tree layers. Each layer was compacted with 25 strokes of the compacting rod. The cone was removed from the concrete by raising it in a vertical direction in time to 5 s. The operation from the start of filling to the removal of the cone was made in time not exceeding 130 s. The high of the slump was measured and compare with the high of the cone immediately after its removal. In each concrete series, the slump was stable after cone removal.

### 3.3. Testing of Compressive Strength of Concrete

Test to determine uniaxial compressive strength were carried out on cubic samples of 100 mm side dimensions using a MEGA 6-3000-150 (PW-MEGA) automatic hydraulic press with a maximum compressive load of 3000 kN. The samples were tested with a constant rate of loading within a range between 0.5 and 0.7 MPa/s. The initial load was 30 MPa for each tested specimen. Tests to determine compressive strength were carried out for concrete recipes SC1, GC1, GC2, BC1 and BC2 after 2, 7 and 28 days of concrete maturation following the EN 12390–3 standard [32]. Three rectangular specimens in every maturation period were subjected to testing in each series. The samples were stored in water at a temperature of 20–22 °C under the EN 12390–2 standard [33]. 

### 3.4. Testing of Flexural Strength of Concrete

The flexural strength of concrete was tested in a center-point loading scheme following the EN 12390-5 standard [34]. The tests were carried out for the SC1, GC1, GC2, BC1 and BC2 concrete series after 7 and 28 days of maturation. Three rectangular specimens with dimensions of 40 mm × 40 mm × 160 mm were subjected to testing in each series. The maximum force, F, for each specimen was determined experimentally using the flexural analysis equipment of a MEGA 6-3000-150 (PW-MEGA) hydraulic press with a maximum flexural load of 150 kN. The specimens were tested with a constant rate of loading of 0.18 kN/s, and the initial load was 0.5 kN for each tested specimen. The flexural strength of concrete was calculated from the following equation:(1)$$f_{cf}=\frac{3Fl}{2d_{1}d_{2}^{2}}$$
where:
*f_cf_*—the flexural strength of the concrete (MPa);*F*—the maximum load (N);*l =* 120—the distance between the supporting rollers (mm);*d*_1_ = 40, *d*_2_ = 40—the lateral dimensions of the specimen (mm).

### 3.5. Water Absorption by Hardened Concrete

The water absorption by hardened concrete was investigated by comparing the mass of samples before insertion into the water and instantly after their removal from the water. The tests were carried out on three cubic samples of 100 mm side dimensions for each of the concrete series of SC1, GC1, GC2, BC1 and BC2. After being cured for 20 h, the concrete samples were weighed and placed in water at a temperature of 20 °C. For 28 days, the concrete samples were conditioned in water at a temperature of 20–22 °C. To provide the proper water circulation, a minimum of 20 mm of space was left between the samples, and between the surface of the curing tank. After 28 days, the specimens were removed from the water and reweighed. The measurements of the samples’ masses were made with an accuracy of 0.01 g using an electronic balance. The parameter of water absorbability by hardened concrete was determined under the formula:(2)$$w_{i}=\frac{m_{i}^{28}-m_{i}^{1}}{m_{i}^{1}}\cdot 100$$
where:
*w_i_*—the parameter of water absorbability by hardened concrete in the *i*-th sample (%);$m_{i}^{1}$ —the mass of the *i*-th sample after 20 h of curing (g);$m_{i}^{28}$ —the mass of the *i*-th sample after 28 days of curing (g).

## 4. Results

### 4.1. Concrete Mix Consistency

The results presented in Table 3 display the effect of the type of aggregate on the change in the consistency of concrete mixes. The S3 consistency class was supported in all of the concrete mixtures analyzed. However, SC1 concrete based on sand had the highest flowability. The use of basalt aggregate resulted in a reduction in the flowability of concrete mixes. The cone falls of BC1 and BC2 concrete mixes was 10 and 20 mm less than SC1, respectively. The lowest cone fall was obtained for GC1 and GC2 concretes based on granite aggregate.

### 4.2. Compressive Strength of Concrete

The results of the compressive strength tests are shown in Figure 1. The highest compressive strength was obtained for the BC1 and BC2 concretes containing basalt aggregate. The early strength of the BC2 concrete on the second day of maturation was 5% higher than the strength of the BC1 concrete, 23% higher than the strength of the SC1 concrete and 71% higher than the strength of the GC1 concrete. It should be noted that the GC1 concrete was an equivalent concrete made with granite aggregate. Simultaneously, the compressive strength of the BC1 concrete was 21% higher compared with its counterpart made with granite aggregate, i.e., the GC2 concrete. After 7 days of maturation, the strengths of the GC1, GC2 and SC1 concretes were very similar. The highest strength value was obtained for the BC1 concrete series, which was 5% higher than the strength of BC2. The compressive strength of the BC2 concrete was 12% higher than that of the GC1 concrete and 13% higher than that of the SC1 concrete. After 28 days, concrete from the BC1 series was characterized by the highest strength, i.e., 150.23 MPa. It was 1% higher than the strength of the BC2 concrete, and 26% higher than the strength of GC2. The strength of the BC2 concrete was 17% higher than the strength of the GC1 concrete and 28% higher than the strength of SC1 concrete. The fracture mechanism of the testing specimens of the SC1, GC1 and GC2 series were very similar. Under the maximal load, all four exposed surfaces were similarly damaged. They were all detached from the rest of the specimen in the same way, while the surfaces in contact with the platens only displayed a little damage at the edges. The same fracture mechanism was observed for specimens of the BC1 and BC2 series after 2 and 7 days of maturation. After 28 days of maturation, explosive failure was observed for the BC1 and BC2concrete specimens.

The impact of aggregate type on the development of compressive strength in relation to the maturation time of the concrete was also analyzed. Figure 2 presents changes in the compressive strengths of concrete concerning the strengths obtained on the 28th day of maturation (*f_cm_*_,28_), which was determined to be 100%. After two days of maturation, the compressive strength of the SC1 concrete was 71% *f_ctm_*_,28__,_ while for the BC2 series concrete, it was 69% *f_cm_*_,28_, and for the GC1 concrete, it was only 47% *f_cm_*_,28_. After 7 days of maturation, the proportions of compressive strength were similar for all the series of concrete. The strength of the SC1 concrete was 82% *f_cm_*_,28_, and the strength of the GC2 concrete was 80% *f_cm_*_,28_. The proportion for the GC1 and BC1 concretes was the same, which was 76%, while the lowest value was found for BC2 concrete, which was 73%.

### 4.3. Flexural Strength of Concrete

Figure 3 and Table 4 show the results of concrete flexural strength testing after 7 days of maturation. The highest value of flexural strength was recorded for the BC2 and BC1 concrete series. The strength of the GC1 concrete was 9% lower than the strength of BC2 concrete. Compared to the results obtained for the BC1 concrete, the strength of GC1 concrete was 20% lower, while the strength of the SC1 concrete was 37% lower.

Figure 4 and Table 5 summarize the results of concrete flexural strength testing after 28 days of maturation. After 28 days, as after 7 days, the highest strength was recorded for concrete in the BC2 series. The flexural strength of the BC1 concrete was 4% lower than the maximal flexural strength of the BC2 concrete. Compared to the BC2 concrete, the flexural strength of GC1 was 10% lower, and the flexural strength of SC1 concrete was 17% lower. The flexural strength of theGC2 concrete was 4% lower than that recorded for the BC1 concrete.

Figure 5 presents the changes in the mean values of the flexural strength of concrete in relation to the strength obtained on the 28th day of maturation (*f_cfm_*_,28_), which was determined to be 100%. On the 7th day of maturation, the flexural strength of the BC1 concrete was 79% *f_cfm_*_,28_, and the flexural strengths of the BC2 and GC2 concrete were 76% *f_cfm_*_,28_. The greatest decrease in the flexural strength after early maturation was observed in the SC1 concrete, for which the flexural strength was 66% *f_cfm_*_,28_.

Figure 6 shows the relationship between the compressive and the flexural strength of concrete after 28 days. The highest $\alpha_{cf}=\frac{f_{cfm}}{f_{cm}}$ strength ductility coefficient value of *α_cf_* = 0.145 characterized the GC2 concrete. For the SC1 concrete, the value of the coefficient was *α_cf_* = 0.137. The lowest values of the *α_cf_* coefficient were achieved for concretes based on basalt aggregate, i.e., BC1 and BC2, the values of which were *α_cf_* = 0.119 and *α_cf_* = 0.126, respectively.

### 4.4. Water Absorption

Figure 7 summarize the results of the test of water absorption by hardened concrete for each series, after 28 days of maturation. The SC1 concrete was characterized by the lowest water absorption rate. Concretes made using basalt aggregate, i.e., BC1 and BC2 were characterized by 23% and 25% higher water absorption rates, respectively. The highest water absorption rate was found in concretes made from granite aggregate. The water absorption rate of the GC1 concrete was 61% higher than that of the SC1 concrete and 29% higher than that of the BC2 concrete.

## 5. Discussion

The results of the tests conducted indicate that it is possible to create high-strength concrete while maintaining the S3 consistency class allowing its use in structures containing spatial reinforcement. When the concrete mix consistency was tested, the SC1 concrete made from sand was characterized by the highest degree of plasticization. The use of basalt aggregate reduced the plasticity of the concrete mix. However, the greatest reduction in plasticity was observed after the use of granite aggregate. Comparing the cone fall obtained for the GC1 and GC2 concrete series, it was found that with an increasing amount of aggregate in the concrete mix, its consistency became less plastic. A similar phenomenon was observed in the case of concretes made of basalt aggregate.

The analysis of the test results shows that the type of aggregate used had a significant influence of on the compressive strength of concrete. Base concrete SC1 made from sand had the lowest compressive strength after 28 days. Replacing sand with granite aggregate in the GC1 series concrete resulted in an increase in compressive strength by 9%, while the use of basalt aggregate increased the strength of theBC2 concrete by 28%. The influence of the type of aggregate was also visible when changing its amount in the concrete mix. Concerning concretes made with granite aggregate, concrete from the GC1 series had 7% higher compressive strength after 28 days than the GC2 concrete containing an increased proportion of aggregate, while in the BC1 and BC2 concretes, which were made from basalt aggregate, there was only a 1% Difference. The test also showed that the most significant delay in the early-age compressive strength was on GC1 concrete made from the granite aggregate. This phenomenon was particularly evident on the second day of concrete maturation.

Concrete based on basalt aggregate had the highest flexural strength after both 7 and 28 days, while the concrete made from sand had the lowest flexural strength value. Comparing the flexural strength to the compressive strength, the highest value was obtained in concretes made from granite aggregate. The *α_cf_* coefficient can be treated as a measure of the ductility of concrete. Similar coefficient was used by Liu et al. in [35] and by Wang et al. [36]. An increased value of the *α_cf_* coefficient in concretes made from granite and sand indicated the better adhesion of these aggregates to the cement matrix. For concretes made from basalt aggregate, this coefficient was the smallest among the series of concrete mixes analyzed.

The type of aggregate significantly affects the water absorption by hardened concrete. Concretes made from basalt aggregate were characterized by water absorption rates more than 20% higher than those made from the sand. The highest water absorption rate was found in concretes made from granite aggregate.

Although the concrete mixes contained a significant amount of Portland cement, their very high strength properties allowed for a considerable reduction in the cross-sectional dimensions and mass of structural elements, which will lead to a reduction in the carbon footprint.

A comparative analysis of the presented solutions was carried out with the solutions found in the literature. The analyzed concrete mixes were made with the use of CEM I 52.5 R Portland cement. The composition of the MC1 (HPC) concrete mix presented by Máca et al. in [6] was compared to the composition of our own mixture, SC1, which is shown in Table 6. The MC1 (HPC) concrete mix was made of sand, microsilica and glass powder. The consistency class measured using the slump test was defined as S3. The composition of the MC1 (HPC) concrete mix had 44% more cement compared to the SC1 concrete and more than twice the amount of microsilica. The amount of sand in the MC1 (HPC) concrete was 13% lower than in SC1 concrete; however, it was sand with a fraction of 0.1–0.8 mm. The water content was almost the same, but the amount of superplasticizers was more than five times greater than in the SC1 concrete. The comparative analysis also included the MC2 (3-900) concrete mix presented by Li et al. in [22], and it was compared to our own recipe, BC2 concrete. The MC2 (3-900) concrete mix was based on sand, basalt aggregate and limestone powder with the addition of microsilica. Additionally, 22% more cement was used in the MC2 (3-900) concrete mix than in the BC2 concrete, while the amount of microsilica was reduced by 39%. In the MC2 (3-900) concrete mix, sand was used as a fine aggregate, which was 7.6% more than the amount of 0.125–1.0 basalt aggregate used in the BC2 concrete. Simultaneously in the MC2 (3-900) concrete mix, 1–3 mm of basalt aggregate was used, which was 15% less than the 1–2 mm aggregate used in the BC2 concrete.

Figure 8 shows a comparison of the experimental results of the compressive strength of concrete after 28 days of maturation. The tests on the MC2 (3-900) concrete, similarly to SC1 and BC2, were carried out on cubic samples with dimensions of 100 mm. MC1 (HPC) concrete tests were carried out on cylindrical specimens with diameters of 100 mm and heights of 200 mm. Following the paper of Siwiński et al. [30], for high-strength concretes, the difference in the compressive strength tested on cubic samples and cylindrical samples with the given dimensions did not exceed 2%; therefore, the results were compared directly. Additionally, in Figure 8 the red color shows the mass of cement used in the preparation of 1 m^3^ of concrete in each series.

The compressive strength of the MC1 (HPC) concrete was 14% higher than that of the SC1 concrete, but it required the use of 246 kg more cement for 1 m^3^ of concrete mix. The BC2 concrete is characterized by the highest compressive strength among the concrete mixes analyzed. Its strength was 3% higher than the compressive strength of the MC2 (3-900) concrete; however, it requires the use of 121 kg less cement per 1 m^3^ of concrete.

The following features distinguish the presented concretes in comparison with conventional concretes:Low cement consumption;No need to use special curing conditions;Use of waste additives such as fly ash;The consistency that enables the use of concrete in an element with a complex reinforcement system.

## 6. Conclusions

The following conclusions can be formulated from the research conducted: The use of basalt aggregate in a concrete mix made with CEM I 52.5 R Portland cement below 600 kg/m^3^ and with the addition of microsilica and fly ash allowed concrete with a compressive strength exceeding 150 MPa, while maintaining S3 consistency to be obtaining. Obtaining such a level of compressive strength in concrete did not require special care conditions or the addition of fibers.The water demand of an aggregate and its quantity affected the consistency of a concrete mix. Increasing the amount of crushed aggregate in a concrete mix reduced its plasticity. In the recipe obtained for very-high-strength concrete based on sand or granite as a natural fine aggregate, changing the aggregate to basalt increased the compressive strength of concrete by 28%, which allowed it to be classified as ultra-high-strength concrete.Concrete made from sand had the lowest compressive strength among all of the concrete mixes analyzed. The achieved strength allowed it to be classified as very-high-strength concrete. The SC1 concrete mix was characterized by the highest degree of plasticization. The flexural strength was almost 14% of the compressive strength of the concrete. This concrete was characterized by the lowest water absorption rate.Concrete made based on granite aggregate was characterized by the lowest degree of mix liquefaction and the lowest early-age compressive strength. However, after 28 days, the GC1 concrete had a compressive strength 9% higher than that of the SC1 concrete. The flexural strength of the GC1 concrete was 13% of the compressive strength, and of the GC2 concrete—of 14.5%. This concrete was characterized by the highest water absorption rate, which was caused by the absorption of granite aggregate.Concrete made from basalt aggregate was characterized by the highest strength throughout the entire maturation period. This applied to both compressive and flexural strength. The strength of the BC1 concrete allowed it to be classified as ultra-high-strength concrete. The liquefaction of the mix was slightly lower than the consistency of the concrete mix made from sand. Water absorption rate was greater than that of concrete made from sand by no more than 25%.

## Figures and Tables

**Figure 1 materials-15-05072-f001:**
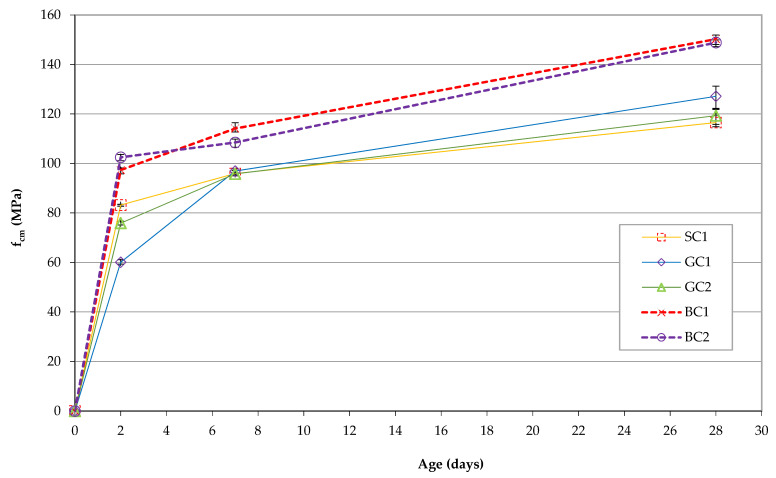
Compressive strength of the analyzed concretes as a function of maturation time.

**Figure 2 materials-15-05072-f002:**
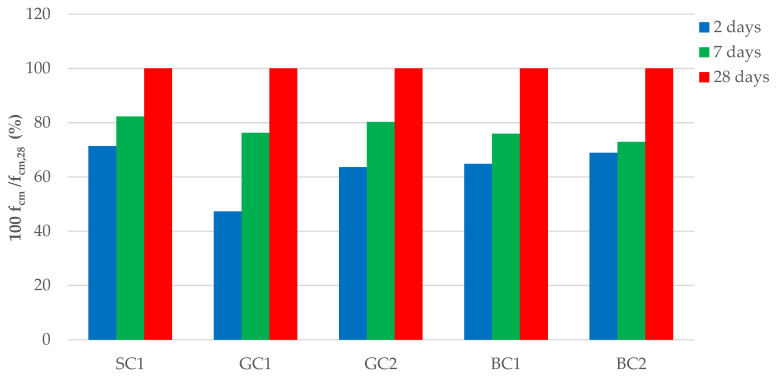
Relative changes in the compressive strength of concretes in subsequent maturing periods with compressive strength after 28 days.

**Figure 3 materials-15-05072-f003:**
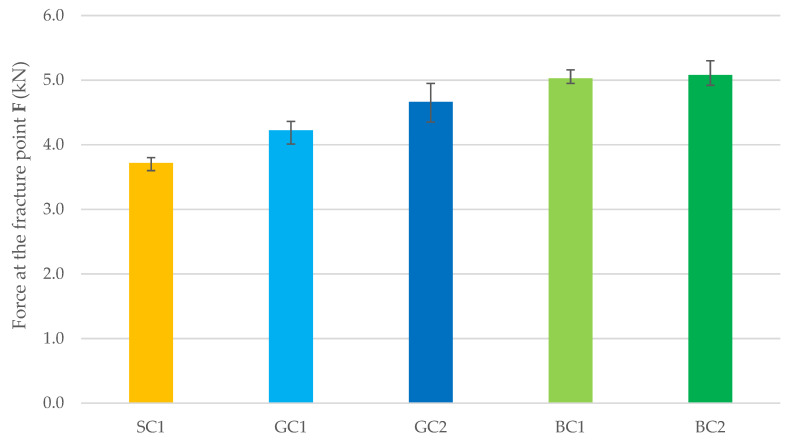
Force at the fracture point for the tested mix series after 7 days of concrete maturation.

**Figure 4 materials-15-05072-f004:**
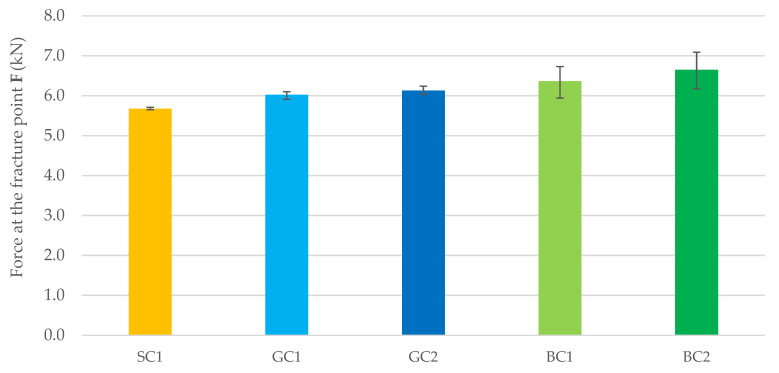
Force at the fracture point for the tested mix series after 28 days of concrete maturation.

**Figure 5 materials-15-05072-f005:**
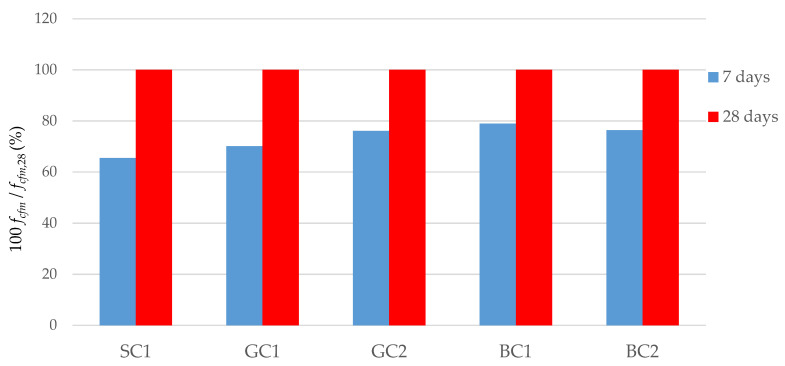
Relative changes in the flexural strength of concretes.

**Figure 6 materials-15-05072-f006:**
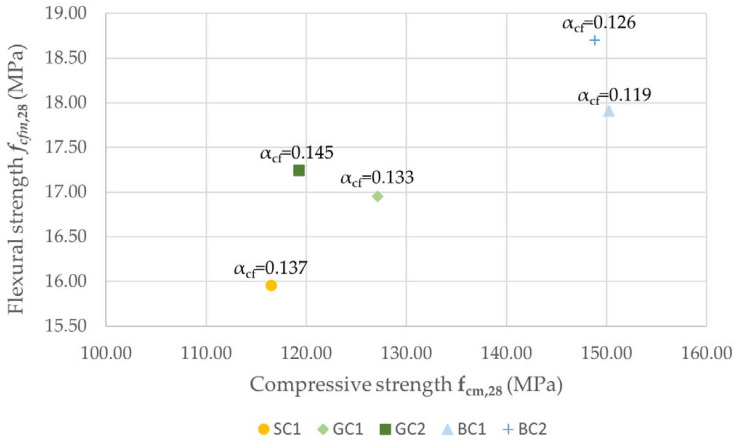
Compressive strength of concretes related to flexural strength after 28 days.

**Figure 7 materials-15-05072-f007:**
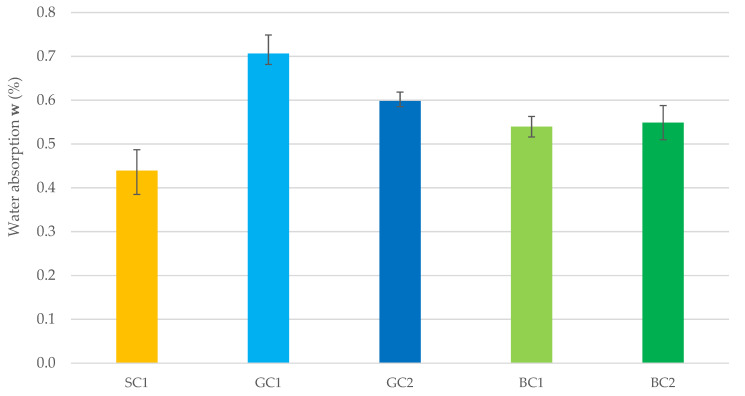
The water absorption by hardened concrete after 28 days for the tested mix series.

**Figure 8 materials-15-05072-f008:**
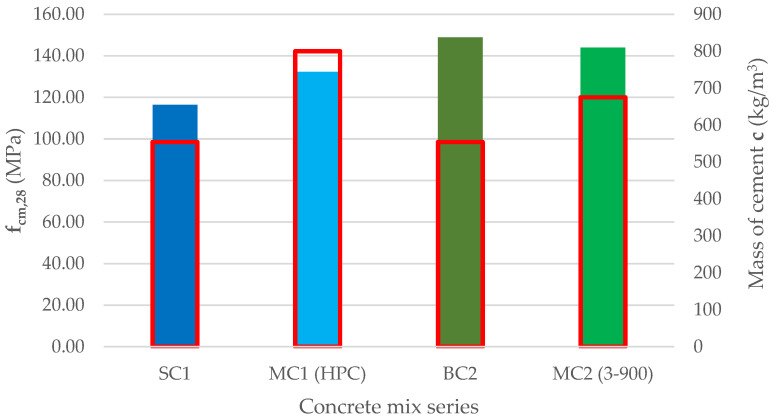
Mean compressive strength values after 28 days of maturation and amount of cement for the analyzed concrete mix series.

**Table 1 materials-15-05072-t001:** Chemical composition of cement, microsilica and fly ash.

Component	Cement (Mass %)	Microsilica (Mass %)	Fly Ash (Mass %)
CaO	67.42	0.10	2.77
SiO_2_	22.56	96.20	51.20
Al_2_O_3_	2.69	-	30.26
SO_3_	2.10	0.20	0.40
Fe_2_O_3_	0.19	0.50	5.36
K_2_O	0.03	1.30	2.64
MgO	-	1.70	1.84
Na_2_O	-	-	0.52

**Table 2 materials-15-05072-t002:** Components of concrete mixes, w/c and w/b ratio.

Component	Sand		Granite				Basalt			
	SC1		GC1		GC2		BC1		BC2	
	kg/m^3^	% c	kg/m^3^	% c	kg/m^3^	% c	kg/m^3^	% c	kg/m^3^	% c
Cement (c)	554	100.0	554	100.0	520	100.0	589	100.0	554	100.0
Fly ash (fa)	92	16.7	92	16.7	87	16.7	98	16.7	92	16.7
Microsilica fume (ms)	74	13.3	74	13.3	69	13.3	79	13.3	74	13.3
Aggregate 0.125–1.0 mm	710	128.1	710	128.1	754	145.0	754	128.1	803	145.0
Aggregate 1–2 mm	600	108.3	600	108.3	638	122.6	638	108.3	679	122.6
Water (w)	175	31.7	175	31.7	165	31.7	187	31.7	176	31.7
Admixture	7.2	1.3	7.2	1.3	8.1	1.6	7.7	1.5	8.6	1.6
w/c					0.33					
w/b					0.26					

**Table 3 materials-15-05072-t003:** Summary of the results of the concrete mix consistency tests.

Concrete Mix Series	Cone Fall (mm)	Class of Consistency
SC1	140	S3
GC1	120	S3
GC2	110	S3
BC1	130	S3
BC2	120	S3

**Table 4 materials-15-05072-t004:** Flexural strength of concrete after 7 days.

Concrete Mix Series	Flexural Strength *f_cfi_* (MPa)			Flexural Strength Mean Value *f_cfm_* (MPa)	Flexural Strength Root Mean Square Error RMSE (MPa)
	*i* = 1	*i* = 2	*i* = 3		
SC1	10.69	10.13	10.55	10.45	0.14
GC1	12.09	11.28	12.26	11.88	0.25
GC2	12.23	13.92	13.19	13.12	0.40
BC1	13.98	13.92	14.51	14.14	0.15
BC2	14.12	14.91	13.84	14.29	0.26

**Table 5 materials-15-05072-t005:** Flexural strength of concrete after 28 days.

Concrete Mix Series	Flexural Strength *f_cfi_* (MPa)			Flexural Strength Mean Value *f_cfm_* (MPa)	Flexural Strength Root Mean Square Error RMSE (MPa)
	*i* = 1	*i* = 2	*i* = 3		
SC1	15.92	16.06	15.89	15.96	0.04
GC1	17.16	17.07	16.62	16.95	0.14
GC2	16.99	17.18	17.55	17.24	0.13
BC1	18.93	16.71	18.08	17.91	0.53
BC2	19.94	17.35	18.82	18.70	0.61

**Table 6 materials-15-05072-t006:** Components of concrete mixes taken from literature compared to SC1 and BC2 concrete mixes.

Component	MC1(HPC)		MC1(HPC)/SC1	MC2(3-900)		MC2(3-900)/BC2
	kg/m^3^	% c	kg/m^3^	kg/m^3^	% c	kg/m^3^
Cement CEM I 52.5 R	800.0	100.0	↑ 44 %	675.0	100.0	↑ 22 %
Microsilica	200.0	25.0	↑ 171%	45	6.7	↓ 39 %
Limestone powder	0	0	-	180.0	26.7	-
Glass powder	200.0	25.0	-	0	0	-
Sand 0–2 mm	1136.0 ^1^	142.0	↓ 13 %	864.5	128.1	↑ 7.6 % ^2^
Basalt 1–3 mm	0	0	-	576.3	85.4	↓ 15 % ^3^
Water	176.0	22.0	↑ 0.3 %	180.0	26.7	↑ 2.3 %
Admixtures	40.0	5.0	↑ 456 %	10.8	1.6	↑ 26 %
w/c	0.27		↓ 18 %	0.28		↓ 15 %

^1^ Fine sand 0.1–0.8 mm, ^2^ amount of sand was compared with amount of basalt aggregate 0.125–1.0 mm, ^3^ amount of basalt aggregate 1–3 mm was compared with amount of basalt aggregate 1–2 mm.

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
