# Peer review of "Effect of Aggregate Type on Properties of Ultra-High-Strength Concrete"

_materials, 2022, doi:10.3390/ma15145072_

Round 1

Reviewer 1 Report

This work presents an analysis of natural fine aggregates' influence on properties of the ultra-high-strength concrete, taking as reference concrete mix is made of natural sand with the addition of fly as hand micro silica, and avoiding the traditional method of the addition of fibers.

The reference concrete mix was modified by swapping sand with granite and basalt aggregate in the same fractions, which modified de standard route. Five series of concrete mixes made with CEM I 52.5R cement were evaluated: consistency of the concrete mix, compressive strength, flexural strength, and water absorption.

The research has been conducted with an appropriate methodology and the results have been presented and discussed clearly, resulting in a comparative analysis of the obtained results indicating a  significant improvement of the concrete strength after the use of basalt aggregate.

This paper’s results show the considerable influence of the type of used aggregate on the mechanical and physical properties of ultra-high high-strength concrete, which is technically interesting for engineering applications.

Additional comments:

·         In-depth revision of the grammar and wording of the text by an expert in English.

·         DO NOT mix British and US English expressions.

·         Line 188: d1 and d2 should be identified as the two measures are not interchangeable for the calculation of resistance.

·         Most of the comments are included in the original text document of the article with change control.

·         Include the characteristics and models of the equipment and infrastructure used to conduct the experimental analysis.

·         Indicate how tests of each condition have been conducted.

·         Include in all graphs and tables the mean square error.

·         Do not use the standard deviation as the statistical parameter of the error; this is incorrect. Use the root mean square error, which is what the error theory indicates. For easier reading of the results, the values should be presented as Mean value ± Root mean square error.

Reviewer 2 Report

The paper investigate the effect of aggregate type on properties of ultra-high-strength concrete. In the review, I think this paper should make some revision.

1. The mechanical testing method should give the standard criteria and give the description of the specimen and facility. 

2. The author should give some explanation on the difference from other concrete.

3. The author should give the fracture feature for the testing specimen.

4. Why did the author select the percentage of concrete? If possible, it should be compare with other commercial concrete.

5.How did the author define the water absorption by hardened concrete? Could give some references?

Reviewer 3 Report

Please see the attached file for detailed comments. I have three major concerns with the study

1) the experimental methods used in the study are not described in sufficient detail to allow other researchers around the world to replicate the study if they choose to do so.

2) mixture design is provided as weight ratio instead of volumetric mixture design. As different aggregates (which have different densities) are used to prepare concrete, each concrete will have different volume of aggregate and paste. Therefore comparison between different mixtures may not be justified. This needs to be corrected prior to publication.

3) Same data is repeated in figure and tables. You need to minimize repetition. if the data is presented in a figure, table should be avoided. If the data is presented in table, it should not be repeated in a figure. Please use figures and tables more efficiently in the revised manuscript.

Detailed comments can be observed for highlighted text in the attached file.

Good luck! 

Reviewer 4 Report

The following queries to be addressed to strengthen the paper:

What will be the application of this work and its advantages compared to conventional ultra high performance concrete?

Ultra high strength means the compressive strength should be more than 150 MPa as well as w/c ratio should be less than 0.20 and binder content will be more than 800 kg/m3. But , these conditions were not satisfied in this work.

Size of the specimens taken as 100 mm cube might be larger than the conventional compressive strength test for ultra high performance concrete.

What is S3? Any standards? If so, what will be the range of cone fall?

Justification for the variation in the test results are lagging.

Microstructural studies could have done to justify the test results.

cross-referencing is missing.

Round 2

Reviewer 1 Report

The paper is ready to be published.

Reviewer 2 Report

All comments had modified and replied. The paper could be accepted as this revised form.

Reviewer 4 Report

Comments were carried out with proper response.